# miR-218 Regulates the Excitability of VTA Dopamine Neurons and the Mesoaccumbens Pathway in Mice

**DOI:** 10.3390/brainsci15101080

**Published:** 2025-10-06

**Authors:** Salvatore Pulcrano, Sebastian L. D’Addario, Mauro Federici, Nicola B. Mercuri, Patrizia Longone, Gian Carlo Bellenchi, Ezia Guatteo

**Affiliations:** 1Santa Lucia Foundation IRCCS, Experimental Neurology Laboratory, Via del Fosso di Fiorano 64, 00143 Rome, Italy; pulcrano.salv@gmail.com (S.P.); daddariosebastianluca@gmail.com (S.L.D.); m.federici@hsantalucia.it (M.F.); mercurin@med.uniroma2.it (N.B.M.); giancarlo.bellenchi@gmail.com (G.C.B.); 2Institute of Biochemistry and Cell Biology IBBC-CNR, Via Pietro Castellino 111, 80131 Naples, Italy; 3Department of Systems Medicine, University of Rome Tor Vergata, Via Montpellier 1, 00133 Rome, Italy; 4Santa Lucia Foundation IRCCS, Molecular Neurobiology Laboratory, Via del Fosso di Fiorano 64, 00143 Rome, Italy; p.longone@hsantalucia.it; 5Department of Medical, Human Movement and Well-Being Sciences, University of Naples ‘Parthenope’, Via Medina 40, 80133 Naples, Italy

**Keywords:** ventral tegmental area, dopamine neuron maturation, miR-218, neuronal excitability, gene expression regulation, behavior, dopamine release, amperometry

## Abstract

Background. MiR-218 is a micro-RNA expressed in two isoforms (miR-218-1 and miR-218-2) in the brain and, within the mesencephalic area, it represents a specific regulator of differentiation and functional maturation of the dopamine-releasing neurons (DAn). Deletion of miR-218 isoforms within the midbrain alters the expression of synaptic mRNAs, the neuronal excitability of DAn of the substantia nigra pars compacta (SNpc), and their ability to release dopamine (DA) within the dorsal striatum. Objectives. Here we have investigated if miR-218 impacts the function of the DAn population adjacent to SNpc, the mesencephalic ventral tegmental area (VTA) innervating the nucleus accumbens (NAcc), and the medial prefrontal cortex. Methods. With the use of miR-218-1, miR-218-2, and double conditional knock-out mice (KO1, c-KO2, c-dKO), we performed electrophysiological recordings in VTA DAn to investigate firing activity, measurements of DA release in NAcc slices by constant potential amperometry (CPA), and in vivo behavioral analysis. Results. We find that KO1 VTA neurons display hyperexcitability in comparison with c-KO2, c-dKO, and wild type (WT) neurons. DA efflux in the NAcc core and shell is reduced in all single- and double-conditional KO striatal slices in comparison with controls. The KO1 mice display a tendency toward an anxiety-like trait, as revealed by the elevated plus maze test. Conclusions. Our data indicate that miR-218-1 is the isoform that mainly regulates VTA DA neuron excitability whereas both miR-218-1 and miR-218-2 impair DA release in the mesoaccumbens pathway.

## 1. Introduction

The substantia nigra pars compacta (SNpc) and ventral tegmental area (VTA) are two adjacent dopaminergic nuclei within the mesencephalon that represent the main source of dopamine (DA) in the brain. Despite overt similarities between the two neuronal populations, such as their low-frequency pacemaker firing, expression of a typical hyperpolarization-activated membrane current, and ability to synthetize and release dopamine in overlapping projection brain areas [1,2,3], they appear metabolically different [4] and display different susceptibility to neuronal demise in Parkinson’s disease (PD) patients and PD models, with SNpc DAn displaying higher vulnerability over VTA DAn [5,6,7]. Dopaminergic fibers arising from the SNpc give rise to the nigrostriatal pathway, which provides dense innervation of the dorsolateral striatum. In parallel, projections from the VTA target the nucleus accumbens (NAcc) and the medial prefrontal cortex (mPFC), forming the mesolimbic (or mesoaccumbens) and mesocortical pathways, respectively. The substantia nigra-dorsolateral striatum pathway is classically implicated in the control of voluntary movement by releasing DA in the cortico–basal ganglia–thalamic circuitry where it differently modulates the direct and the indirect pathways, with the overall result of facilitating movement [8,9,10]. Thus, the basal ganglia nuclei help to plan and program movements for the primary motor cortex and the spinal cord that have more direct access to the somatic muscles [11]; and dysfunctions in this loop, due to low levels of DA released by nigrostriatal pathway, are associated with movement disorders, such as PD. The mesoaccumbens dopaminergic projection is mainly involved in controlling affective states and behavior, together with hippocampal and basolateral amygdala (BLA) inputs to NAcc [12]. Dopamine release within this circuit is responsive to both reinforcing [13,14] and aversive stimuli [15,16]. Mesoaccumbens and mesocortical projections are also important for reward-driven behaviors [17,18] and altered dopaminergic activity within the VTA might in part underlie anxiety symptoms [19,20] and related disturbances in intrinsic and extrinsic motivation [21,22,23]. The recent literature indicates that a specific VTA dopaminergic circuit projecting to the BLA selectively controls anxiety in mice [24].

MiR-218 is a microRNA present in two isoforms, miR-218-1 and miR-218-2, which regulate the function and morphology of several neuronal subtypes, including spinal motor neurons, hippocampal neurons [25,26], and mesencephalic dopaminergic neurons (DAn; [27,28,29]). With regard to DAn, miR-218 promotes dopaminergic differentiation and innervation of target areas by regulating a group of genes controlling synaptic functions [28,29,30]. Deletion of single miR-218 isoforms in mice (miR-218-1, KO1; miR-218-2, c-KO2) or both (double miR-218 KO, c-dKO) alters the excitability of SNpc DAn, impairs synaptic DA efflux in the dorsolateral striatum, and reduces a Ca^2+^-activated potassium conductance underlying the action potential after-hyperpolarization phase (I_AHP_) [29]. With regard to VTA DAn, miR-218 regulates the expression of the Netrin-1 receptor DCC (deleted in colorectal cancer), which is a factor important for organizing dopaminergic connectivity in the mPFC [28]. However, if miR-218 regulates VTA DA neuronal firing, DA efflux in the target areas and behavior is not known. With the aim to fill the gap, we evaluated the excitability of VTA DAn and measured the amount of synaptic DA release in the NAcc of KO1, c-KO2, c-dKO, and WT mouse brain slices. While DA release is impaired in all KO ventral striatal slices, intrinsic VTA DAn firing appears mainly regulated by the miR-218-1 isoform. Indeed, deletion of miR-218-1 caused hyperexcitability of VTA DAn. Such functional alteration of VTA DAn seems to be associated only with a tendency toward the anxiety-like behavior of KO1 mice.

## 2. Materials and Methods

### 2.1. Animals

The mouse lines miR-218-1^−/−^ (KO1), miR-218-2^−/−^ (c-KO2; see Pulcrano et al., 2023 [29]), miR-218 double KO mice (c-dKO), Tg::TH1/GFP [31], and the En1Cre/1 (kindly provided by Prof. Antonio Simeone) were used for experiments and for breeding [32]. The mouse strains were maintained in an outbred C57Bl/6 background. Wild type, KO1, c-KO2, and c-dKO mice within the age range of P40-P60 were used for electrophysiology and measurements of synaptic DA release. In vivo behavioral analysis was performed on KO1 and WT mice (P40-P60). Both males and females were used, and we found no differences between sexes in all experimental sections. Mice were genotyped as previously described [29].

### 2.2. Ethics Statement

The use of animals for research experiments was in accordance with the guidelines of the Italian Ministry of Health (authorization 968/2023-PR, approved on 14 November 2023 and 1172/2024-PR, approved on 27 December 2024) and with the Italian and European safety and ethical rules and regulations, including the Council Directive 2010/63/UE (published on 22 September 2010) regarding the protection of animals used for experimental and other scientific purposes, as well as national legislations (Italian Legislative Decree 116/92 and Italian Legislative Decree 388/98). We made every effort to reduce or prevent animal suffering and to limit the number of animals used in agreement with the principles of the 3Rs (replacement, reduction, refinement).

### 2.3. Midbrain and Ventral Striatum Slice Preparation

We used halothane to profoundly anesthetize mice by inspiration before decapitation. We quickly pulled out the brain from the skull and separated it into two blocks with a razor blade cut at the level of optic chiasma, with one containing the midbrain and the other containing the cortex and the striatum. The midbrain-containing block was submerged in aCSF at 2–4 °C with the following composition (in mM): NaCl 126, NaHCO_3_ 24, glucose 10, KCl 2.5, CaCl_2_ 2.4, NaH_2_PO_4_ 1.2, and MgCl_2_ 1.2 (95% O_2_–5% CO_2_, pH 7.3), and then glued to a plate of a vibratome (Leica VT 1200 S, Leica Microsystems, Wetzlar, Germany) chamber and sliced horizontally, starting from the ventral surface. Midbrain sections were kept in aCSF at 33.0 ± 0.5 °C for one hour before usage. After 1 h of recovery, one slice was placed in the recording chamber mounted under an up-right microscope (Olympus BX51WI, Tokyo, Japan) and perfused at 2.5–3.0 mL/min with aCSF (33.0 ± 0.5 °C) [3,33,34].

The block containing cortex and striatum was placed in chilled aCSF solution, glued to a plate of a second vibratome (Leica VT 1200 S) and sliced coronally. Cortico–striatal slices (300 µm thickness) were left to recover in a beaker containing aCSF at the temperature of 32 °C, for at least 1 h before usage [3,33,34].

### 2.4. Electrophysiological Recordings in Midbrain Slices

In the recording chamber, a single midbrain slice was held by a U-shaped anchor dipped in flowing aCSF (2.5–3.0 mL/min) at 33.0 ± 0.5 °C. We performed electrophysiological recordings in cell-attached and whole-cell patch-clamp configurations in VTA DA neurons meeting the criteria reported below and according to a map showing VTA DA neuron distribution within midbrain slices from the TH-GFP mice, based on intrinsic properties and DA sensitivity [3].

The following parameters were taken into account to select VTA DA neurons for electrophysiological recordings: (1) location in the lateral VTA, according to [3]; (2) the emission of eGFP fluorescence in response to 480 nm UV light excitation (emission filter, 510 nm) delivered to the slice through an optic fiber connected to a monochromator (Polychrome IV; Till Photonics, Munich, Germany) through a 40× water-immersion objective; a CCD camera (Evolve; Photometrics, Tucson, AZ, USA) installed on the microscope captured the emitted light; (3) the presence of a slow, regular pacemaker firing (0.4–10 Hz) in cell-attached configuration; (4) the presence of a hyperpolarization-activated (I_H_) current in response to hyperpolarizing voltage commands (−60 to −120 mV, 20 mV increment, 1 s) [35]. Recording pipettes were pulled using thin-wall capillaries (WPI) and filled with a solution that contained (in mM) 10 KCl, 125 K-gluconate, 10 HEPES, 0.1 CaCl_2_, 2 MgCl_2_, 4 ATP-Mg_2_, 0.3 GTP-Na_3_, 0.75 EGTA, 10 phospho-creatine-Na_2_, pH 7.2, osmolarity of about 280 mOsm. Cell-attached recordings of spontaneous action potential firing started after a seal resistance > 1 GΩ was established, before rupturing the membrane patch, by means of the Axoscope 9 software (Molecular Devices, San Jose, CA, USA) in the voltage clamp (V_H_ −60 mV; sampling 1 kHz). Measurements of membrane input resistance (R_m_) were obtained through the Clampex 9 Membrane Test protocol, immediately (<2 min) after the establishment of whole cell configuration, that consisted of a 30 ms-long, 5 mV step from V_H_ −60 mV (33.3 Hz). Ten successive traces were averaged to obtain reliable measures.

Current-to-voltage (I/V) relationship of the hyperpolarization-activated current (I_H_) was obtained in voltage clamp recordings by applying 8 voltage steps (−50 to −120 mV, 10 mV increment, 2 s) to VTA DA neurons, V_H_ −50 mV. To evoke action potential firing, a current-clamp protocol was applied to VTA DA neurons, and consisted of a sequence of 26 current steps, each lasting 2 s from 0 to 250 pA (10 pA increments), maintaining membrane potential between the steps at V_H_ −60 mV. Mean instantaneous frequency was calculated by averaging the frequency of the first two action potentials (APs) evoked at each stimulus intensity with a sampling rate of 20 kHz. Rheobase was considered as the minimal depolarizing current amplitude (pA) able to evoke an AP. To measure the current underlying the AP after-hyperpolarization (I_AHP_), we applied a voltage command from a V_H_ of 60 mV to 0 mV, for a duration of 100 ms. We compared the I_AHP_ peak (pA) and area (pA*s) among different genotypes, that were calculated with the Clampfit software, as the maximal current amplitude and the area under the current trace, in a time window frame from the end of the voltage command up to the 2 s time point.

### 2.5. Constant Potential Amperometry (CPA)

Dopamine (DA) release was measured by constant potential amperometry performed as described by Pulcrano et al. (2023, [29]). Through a stimulating bipolar electrode (nickel/chromium-insulated), we delivered electrical stimuli to the neuron axons in the ventral striatum through a DS3 Stimulator (Digitimer, Welwyn Garden City, UK) every 5 min to evoke DA release. Each stimulation was made of a 5-pulse train (250 ms total duration) whose intensity increased progressively (100–600 mA, each pulse lasted 40–60 ms) until reaching a maximal amount of DA release.

Extracellular DA efflux in response to electrical stimulation was recorded as amperometric current, by means of a carbon fiber electrode (30 µm diameter and 100 µm lenght, WPI) which was softly adjusted on the slice surface to a depth of 100–150 µm, in a position close to the stimulating electrode, connected to a potentiostat (MicroC, WPI, Rodheim vor der Höhe, Hessen, Germany). The release of DA was evoked in the nucleus accumbens (NAcc) core and shell, in the medial, lateral, and central parts of the two nuclei. Histograms in Figure 2 were obtained by averaging the values of DA concentration recorded in all observation points (the total number is reported in the figure legend) for each experimental group. The voltage between the carbon fiber electrode and the silver/AgCl pellet was 0.6 V. Amperometric signals were converted into DA concentration at the end of each experiment, by means of a calibration curve obtained by applying three known DA concentrations (1, 3 and 10 µM) and measuring the corresponding amperometric current amplitudes, according to [34].

Amperometric currents were characterized by a fast-rising phase after electrical stimulation and typically returned to baseline within 1.5 s. Amplitudes of amperometric currents were obtained by subtracting from the peak value of the baseline current. Amperometric current latency was calculated as the duration of the time window between the stimulus artifact and the current peak, and the half decay was calculated as the duration of time window between the current peak and time at which it recovered at 50% of maximal amplitude. The signals were digitized through a Digidata 1440A coupled to a personal computer running the Clampex 10 software.

### 2.6. Behavioral Analysis

#### 2.6.1. Elevated Plus Maze Test

Mice were placed in an elevated apparatus (38.5 cm (from the floor) equipped with two open (27 × 5 cm) and two closed (27 × 5 × 15 cm) arms connected by a central plate (5 × 5 cm)). The test lasted 5 min ([36]). We measured the percentage of time spent in the open arms (time in open/open + closed × 100) and data were collected and analyzed by the “EthoVision” (Noldus, Wageningen, The Netherlands).

#### 2.6.2. Marble-Burying Test

A single mouse was positioned in a box (40 × 18 × 26 cm) containing clean home-bedding (5 cm) and 20 marbles (scattered). After 30 min, we counted marbles buried by considering a marble covered ≥50% with bedding as buried [37]. MBT test was performed 48 h after EPM.

### 2.7. Softwares and Statistical Analysis

For statistical analysis, we used the following softwares, the GraphPad Prism 10 (GraphPad Software) and OriginPro 2019 software. Details of the specific tests were used to assess the statistical level of significance between groups or experimental conditions are reported in each figure legend. The Shapiro–Wilk test was used to assess the normality of raw data.

## 3. Results

### 3.1. Isoform miR-218-1 Deletion Alters Firing Properties of VTA DA Neurons

Based on our previous finding, indicating that miR-218 is an important regulator of action potential firing and membrane properties of single DAn of the substantia nigra pars compacta and of the nigro-striatal dopaminergic pathway [29], we aimed at exploring its regulatory action within the dopaminergic nucleus adjacent to SNpc, the VTA, and the mesoaccumbens dopaminergic pathway, which provides DA innervation to the ventral part of the striatum, the nucleus accumbens (NAcc), core, and shell. To this purpose, we performed extracellular single-unit and patch-clamp recordings in the VTA DA neurons in miR-218 KO1, c-KO2, c-dKO, and WT midbrain slices. All animal groups expressed TH-GFP. With regard to passive membrane properties, differently from what we observed in SNpc DA neurons [29], miR-218 deletion does not affect membrane input resistance (R_m_) of VTA DA neurons, and all groups of miR218 KOs displayed similar R_m_ values (Figure 1A) to WT neurons.

By contrast, miR-218 deletion strongly affects active membrane properties. Particularly, the frequency of spontaneous action potential discharge, recorded in the extracellular-single unit to prevent modifications secondary to cytoplasm dialysis, was higher in miR-218 KO1 (4.32 ± 0.9, n = 9 in KO1, vs. 2.06 ± 0.15, n = 11 in WT, Figure 1B, *p* < 0.05), but not in c-KO2 (2.20 ± 0.22, n = 5, *p* = 0.61) nor in c-dKO (3.44 ± 0.98, n = 6, *p* = 0.08) VTA DAn, in comparison with WT DA neurons. Also the evoked firing in response to positive current steps (0–250 pA, 10 pA increase) in whole-cell configuration was modified by miR-218-1 deletion. Indeed, the action potential (AP) instantaneous frequency and maximal number produced during 2 s of stimulus duration were increased only in KO1 VTA DA neurons (Figure 1D), but not in single c-KO2 nor in c-dKO VTA DA neurons. These data were mirrored by a trend toward a slightly lower rheobase, in miR-218 KO1 than in WT DAn (*p* = 0.052, Figure 1C). These results indicate that the miR-218 isoform 1 regulates spontaneous and evoked firing activity in VTA DA neurons, opposite to SNpc DA neurons where the c-dKO genotype displayed the largest functional modification in comparison to controls.

To unravel a possible mechanism underlying changes in VTA DA neurons firing, we investigated two membrane currents known to be characteristic of DA neurons that could affect their firing frequency, namely I_H_ (Figure 1E) and Ca^2+^-activated K^+^ current underlying the action potential after-hyperpolarization (I_AHP_; Figure 1F). We found that I_H_ was significantly reduced, at the more negative membrane potentials, in all KOs VTA DA neurons (Figure 1E), whereas I_AHP_ peak and area were largely unaffected by miR-218 deletion (Figure 1F).

### 3.2. miR-218 Deletion Impairs DA Release in the Nucleus Accumbens

To explore the involvement of miR-218 in synaptic efficacy in the mesoaccumbens pathway, we measured the amount of synaptic DA release within the NAcc core and shell of the ventral striatum, by means of constant potential amperometry (CPA). As previuosly reported [33,34], the recorded amperometric current is mainly caused by the DA released in the extracellular space because the current is not modified by fluoxetine and reboxetine (serotonin and norepinephrine transporter blockers, respectively) [34], suggesting that these neuronal mediators contribute negligibly to the amperometric signal recorded in this brain area.

Electrical stimulation of dopaminergic fibers in the ventral striatal slices of miR-218 KO1, c-KO2, c-dKO, and WT TH-GFP mice, caused DA release which was strongly reduced in all miR-218 KOs, either in the NAcc core (Figure 2A) or shell (Figure 2B). Such reduction was not associated with changes in the latency or duration (half life) of the amperometric current, suggesting normal function of the DA uptake system (largely mediated by the dopamine transporter, DAT) in the NAcc of all KO animals.

### 3.3. Deletion of miR-218-1 Does Not Significantly Affect Anxiety-like Behavior in Mice

Having observed that miR-218-1 is the isoform that mainly affects spontaneous and stimulated action potential firing of VTA DAn and a lower amount of DA is released in the mesoaccumbens pathway in KO1 NAcc, as well as c-KO2 and d-cKO, we next evaluated the impact of miR-218-1 deletion on mice behavior, by exposing KO1 mice to the elevated plus maze (EPM) test and to the marble-burying test in comparison with WT mice (Figure 3).

EPM test revealed that KO1 mice tend to spend less time in the open arms whereas no difference was seen in the number of marbles buried by KO1 with respect to WT mice.

## 4. Discussion

The regulation of gene expression affects multiple cellular processes relevant to health condition and disease, including brain disorders. MicroRNAs are short (20–23 nucleotides in length), single-stranded RNAs that interact with mRNA and regulate their expression by triggering mRNA transcript degradation and translational repression [38,39]. In the brain, a single miRNA is able to control many genes, and genome-wide studies have identified lists of mRNAs whose expression correlate with the translational repression mediated by miRNAs. MiR-218 is specifically involved in the maturation and differentiation of spinal motor neurons, and hippocampal and mesencephalic DA neurons [25,26,27,28]. The manipulation of miR-218 levels has been associated with brain disorders with depression/anxiety traits [40,41]. In line with published evidence indicating that the manipulation of miR-218 expression causes hyperexcitability in other types of neurons [29,42], we found that low expression levels of miR-218 increases excitability of VTA DAn. We highlighted that the regulation of VTA dopaminergic neuron (DAn) function is primarily mediated by the miR-218-1 isoform, as VTA DAn neurons from the miR-218 KO1 mice exhibit clear signs of hyperexcitability, whereas conditional KO2 (c-KO2) and double conditional KO (c-dKO) neurons show electrophysiological properties largely comparable to wild-type (WT). This observation contrasts with what we previously reported in SNpc DAn [29], where the c-dKO condition produced the most severe phenotype, indicating a dosage-dependent effect of miR-218 on SNpc DAn function. Additionally, in VTA DAn I_H_ current appears regulated by miR-218 whereas in SNpc DAn, it is not. The opposite occurs for Ca^2+^-activated K+ currents that are not regulated by miR-218 in VTA but highly dependent on miR-218 expression levels in SNpc DAn [29].

This divergence adds to other well-documented differences between these two dopaminergic populations. Although SNpc and VTA DAn both synthesize and release dopamine into overlapping target areas and share key intrinsic properties—such as low-frequency pacemaker firing and the expression of specific membrane conductances like I_H_ and Ca^2+^-activated potassium currents (I_AHP_) [3]—they differ markedly in their vulnerability to degeneration in PD and related models [5,6,7], possibly due to differing metabolic demands [4].

These functional distinctions suggest that miR-218 isoforms may regulate distinct sets of target genes depending on the neuronal context. In line with this idea, we have previously identified synaptic gene networks regulated by miR-218. Although the mature sequences of the two isoforms are identical, our data suggest they can regulate partially distinct gene targets. Indeed, only a small subset of genes were commonly upregulated in both KO models [29], while a number of others appeared specifically upregulated only in miR-218-1 KO. These include MECP2, Nrxn1, and HCN3—all of which are involved in synaptic function and neuronal excitability. Interestingly, it was recently reported that deletion of Nrxn genes alters the synaptic composition of dopaminergic neurons without affecting their synaptic ultrastructure [43]. Based on these findings, we hypothesized that changes in inhibitory transmission might affect dopamine release. In this context, it is important to note that altered inhibitory transmission has also been observed following miR-218 knockdown in the hippocampus, likely due to the dysregulation of synapse-specific targets [42].

Interestingly, we found that, despite only KO1 VTA DA neurons becoming hyper-excitable, all miR-218 KOs VTA DA neurons display impaired DA release in the NAcc core and shell, indicating that intrinsic excitability is mainly regulated by the miR-218-1 isoform, whereas the machinery of synaptic DA release is controlled by both miR-218 isoforms. These results highlight the concept that a direct correlation between VTA DA neurons firing, recorded at somatic level, and DA release at synaptic terminals in NAcc is not present. A dissociation between DA neuron firing and DA release in NAcc has been previously reported by Mohebi et al. 2019 [44] in which authors described patterns of NAcc dopamine release not generated by VTA DA neurons firing, but rather regulated by firing independent local mechanisms, such as pre-synaptic modulation of dopaminergic terminals by inputs originating in the basolateral amygdala [45] or by cholinergic transmission within the striatum [46]. It is possible that, in the miR-218 KOs model, synaptic DA release might be mainly under local control. Further investigation is needed to ascertain this possibility. Another interpretation of the apparent paradox between KO1 VTA DA neuron hyperexcitability and reduced synaptic DA release in all miR-218 KOs striatal slices may be linked to a technical issue. Indeed, the stimulus intensity delivered to dopaminergic fibers in striatal slices to evoke synaptic DA release is much higher than that used to stimulate somatic action potential firing in the midbrain. Thus, synaptic DA release induced in CPA recordings may be independent of somatic VTA neuron firing and may reflect alterations in the machinery of vesicle release at synaptic terminals of miR-218-deficient mice. We also observed that the reduced DA release is not associated with uptake system alteration, since other parameters of the amperometric current, such as latency and half life, are not affected. We may therefore hypothesize that the observed effects on dopaminergic neuron excitability and dopamine release could result either from intrinsic defects within the dopaminergic neurons (DAn) or from circuit-level alterations following miR-218 deletion. In this case, it would be reasonable to expect that miR-218 deletion may lead to distinct outcomes in the VTA compared to the SNpc, due to their divergent connectivity and microenvironment.

Our behavioral data indicate only a tendency of having an anxiety-like trait in KO1 mice. Additional tests are necessary to unravel miR-218 involvement in mice behavior.

Although our data showing altered intrinsic excitability in iDAn derived from c-dKO miR-218 mice [29] suggest a primary effect on DAn cellular composition rather than a consequence of altered GABAergic innervation, a definitive demonstration would require further investigation. This could be achieved by analyzing the biochemical composition of DAn or by performing cell type-specific deletion of miR-218 in defined neuronal populations. Whatever the underlying mechanism, our data support the idea of isoform-specific contributions by miR-218 to the fine-tuning of dopaminergic neuron physiology.

## 5. Conclusions

In this manuscript, we report that miR-218 is an important regulator of the functional properties of VTA DA neurons. Different isoforms are involved in this process: intrinsic excitability of DA neurons mainly depend on miR-218 isoform 1 whereas evoked synaptic DA release in the NAcc appears to be regulated by both mir-218-1 and miR-218-2. These findings highlight an important difference in the role of miR-218 in regulating VTA DA neuron functions compared with its previously reported role in the adjacent dopaminergic nucleus of the substantia nigra pars compacta. Such brain nucleus-specific regulation by miR-218 is not unexpected, as we have previously reported that each miR-218 isoform affects the expression of specific mRNAs, suggesting the existence of possible regulatory mechanisms that remain to be identified.

## Figures and Tables

**Figure 1 brainsci-15-01080-f001:**
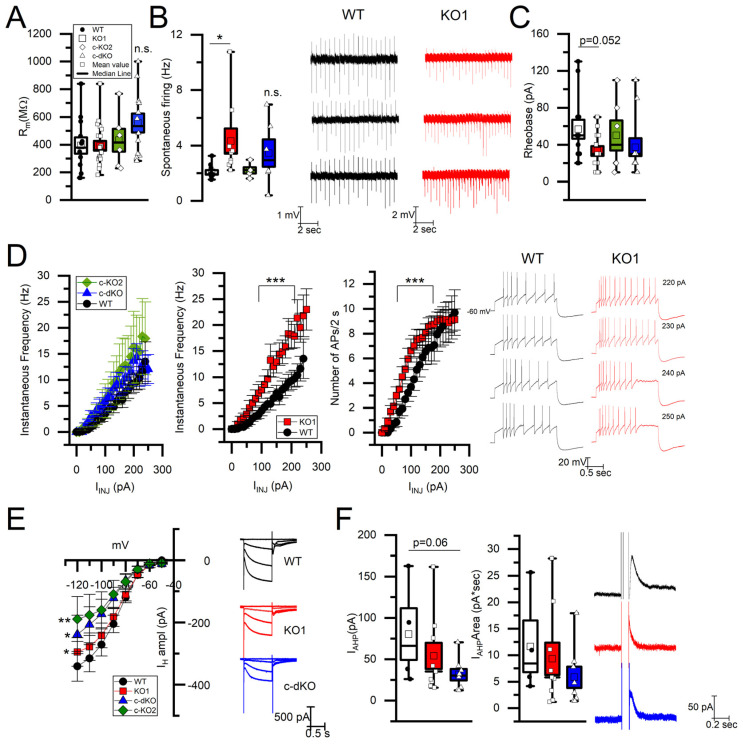
miR-218 KO1 VTA DAn show hyperexcitability. (**A**) Membrane input resistance is similar among KO1 (n = 20), c-KO2 (n = 6), c-dKO (n = 12), and WT (n = 12) VTA DAn (n.s., Student’s *t*-test). (**B**) Box plots and cell-attached traces indicate that the frequency of spontaneous firing is higher selectively in KO1 (n = 9) with respect to WT VTA DAn (n = 11; * *p* < 0.05, Student’s *t*-test) but not in c-KO2 (n = 5) and c-dKO (n = 6) VTA DAn (n.s., Student’s *t*-test). (**C**) Rheobase, the minimum depolarizing current necessary to evoke an action potential, displays a tendency to lower values in KO1 (*p* = 0.052, n = 16, Student’s *t*-test) in comparison with WT (n = 12), but not in c-KO2 (n = 6) nor c-dKO (n = 11) VTA DA neurons. (**D**) Instantaneous frequency and the number of evoked APs by injection of depolarizing current (from 0 to250 pA, with 10 pA increment, 2 s duration) is increased only in KO1 VTA DAn (n = 16, two-way ANOVA, followed by Tukey test for post hoc comparisons, *** *p* < 0.001, 100–200 pA) but not in c-KO2 (n = 6, n.s., two-way ANOVA, Tukey test for post hoc analysis) nor c-dKO (n = 11, n.s. two-way ANOVA, Tukey test for post hoc analysis), in comparison with WT VTA DAn (n = 13). Example traces of APs firing in WT (black) and KO1 (red) VTA DAn at high stimulus intensities (220–250 pA). (**E**) I–V relationship of IH showing reduced amplitudes in all KO groups in comparison with WT VTA DAn (n = 12; KO1, n = 15; c-KO2, n = 6; c-dKO, n = 8; two way ANOVA, Tukey test for post hoc analysis. WT vs. KO1: * *p* < 0.05 at −120 mV; WT vs. c-dKO: * *p* < 0.05 at 110 and −120 mV; WT vs. c-KO2:, ** *p* < 0.01 at −110 and −120 mV). On the right, example traces of IH evoked by voltage steps from −60 to −120 mV, 20 pA increment, VH −60 mV. (**F**) IAHP peak amplitude and area are similar in KO1 (n = 9) and c-dKO (n = 8) in comparison with WT VTA DAn (n = 4, n.s. Student *t*-test). On the right, example traces of IAHP evoked by a voltage step from VH −60 mV to 0 mV, 100 ms. In the box plots, box height indicates S.E.M., in the other plots (line + symbols), data are presented as mean ± S.E.M.

**Figure 2 brainsci-15-01080-f002:**
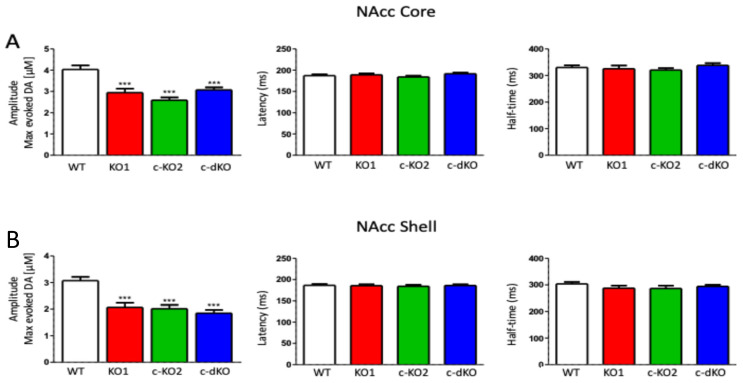
DA release in NAcc core and shell is impaired by miR-218 KO. (**A**) Evoked DA release detected by CPA in NAcc core is decreased in all KOs in comparison with WT striatal slices (n = 40 from five mice; KO1, n = 30 from four mice; c-KO2, n = 52 from four mice; c-dKO, n = 48 from four mice; *** *p* < 0.001, one-way ANOVA, F = 12.77, Bonferroni test for post hoc comparisons). Post hoc analysis revealed no significant differences among KOs. (**B**) Evoked DA release detected by CPA in the NAcc shell (WT, n = 34 from five mice; KO1, n = 25 from four mice; c-KO2, n = 34 from four mice; c-dKO, n = 48 from four mice; *** *p* < 0.001, one-way ANOVA, F = 16.24, Bonferroni test for post hoc comparisons). Post hoc analysis revealed no significant differences among KOs. The reduced amplitude of the amperometric current is not accompanied to changes in the amperometric current latency nor half life (one-way ANOVA, Bonferroni test for post hoc comparisons; F_CORE LATENCY_ = 1.41, F_CORE HALF LIFE_ = 0.9; F_SHELL LATENCY_ = 0.10, F_SHELL HALF LIFE_ = 0.92). Bars indicate means ± S.E.M.

**Figure 3 brainsci-15-01080-f003:**
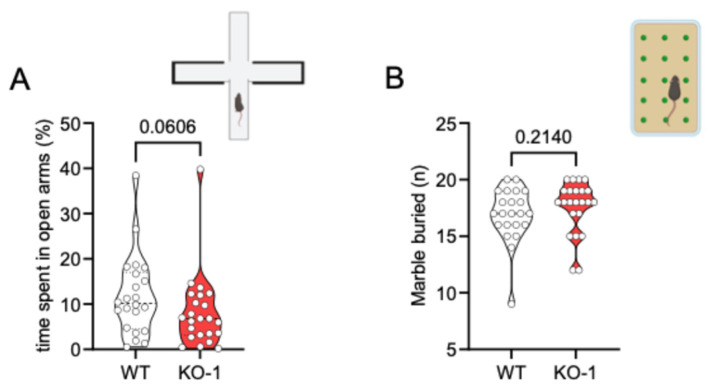
miR-218 KO1 mice show a tendency of anxiety-like behavior. (**A**), Plot indicating the percentage of total time spent in the open arms of the elevated plus maze (EPM) apparatus. (as) KO1 mice display a tendency to spend less time in the open arms (WT n = 22; KO1 n = 23; Mann–Whitney test). (**B**), Number of marbles buried, shows no difference between genotype (WT n = 22; KO1 n = 23, Mann–Whitney test). Data are expressed as median ± quartiles; each dot represents an single animal (the drawings were created with BioRender.com).

## Data Availability

The data presented in this study are available on request from the corresponding author due to lack of time for files preparation.

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
