# Peer review of "miR-218 Regulates the Excitability of VTA Dopamine Neurons and the Mesoaccumbens Pathway in Mice"

_brainsci, 2025, doi:10.3390/brainsci15101080_

Round 1
Reviewer 1 Report
Comments and Suggestions for Authors
Dear authors,
This is an interesting and well-organized study that explores the distinct roles of miR-218 isoforms in VTA dopamine neurons using conditional KO mice, revealing altered excitability, reduced dopamine release in the NAcc, and anxiety-like behavior. The findings highlight miR-218-1 isoform as the predominant regulator of mesoaccumbens dopamine function and associated behavioral traits. I commend the authors for producing a manuscript that is easy to read, well-structured, and engaging, with a clear and thorough presentation of the results, effectively linking the findings to their broader implications in the field.
However, I have few questions and suggestions.
Could the authors clarify how the conclusions of the present study substantially differ from those reported in their previous work - J Neurosci. 2023 Nov 29;43(48):8104-8125. doi: 10.1523/JNEUROSCI.0431-23.2023? What novel insights or original contributions does this study provide beyond the scope of the earlier findings?
Have you considered assessing structural changes in these pathways, such as markers of synaptic plasticity or GABAergic signalling?
Have you considered measuring the levels of DA in these neural correlates?
Have you considered assessing the components of depressive-like behavior?
Reviewer 2 Report
Comments and Suggestions for Authors
This study provides evidence that deletion of miR-218-1, but not miR-218-2, selectively increases the excitability of VTA dopamine neurons and reduces dopamine release in the nucleus accumbens. Behaviourally, miR-218-1 knockout mice display an anxiety-like phenotype in the elevated plus maze. Collectively, these findings identify miR-218-1 as an isoform- and region-specific regulator of mesoaccumbens dopamine circuitry and anxiety-related behaviour.
Major comments
1. Behavioural analysis
The behavioural evidence for an anxiety-like phenotype is relatively limited, being based only on the elevated plus maze and marble burying test. To strengthen the conclusion, it is important to include at least one additional complementary assay (e.g., open field or light–dark box test). Furthermore, potential sex-specific differences in behaviour and physiology should be analysed.
2. Figure presentation
Figure 1B and Figure 1D rely solely on colour for genotype distinction. Differential line styles or symbol coding should be added.
Minor comments
Line 35: “VTA DAn neurons function” → “VTA DA neurons”
Line 44: “such as,their low frequency” → “such as their low-frequency”
Line 241: “an hallmark” → “a hallmark”
Line 280: “a tendency to spent” → “a tendency to spend”
Line 282: “show no difference” → “shows no difference”
Reviewer 3 Report
Comments and Suggestions for Authors The data presented in this paper are not strong enough to convince the Readers about the hypothesis of the Authors. Statistical analysis of the data indicates weak significance, if here is at all, in two sets of experiments, Fig. 1C and Fig. 3A. People making experimental work know it may happen, the only possibility in such case is to repeat the experiments to reach a definitive conclusion. Comments others Indicate the aim of this study in a section added to the end of the Introduction. Indicate the age and weight of the mice used in the experiments. Vatiations observed in the data might be due to the fact that both sexes were involved in the experiments, the Authors refer their own paper here. It would help to have impression about slice preparation if the Authors added microphotograph about the slices prepared (2.3. and 2.4.). 2.7. List all statistical procedures used in this study. Indicate whether S.D. or mean±S.E.M. was calculated. Also add this in figure legends. It is KO-1 in Fig. 1. and is KO1 in other places. Fig. 2. Consider to used one-way ANOV and a post hoc test for data analysis presented here. Data presented in the behavioral tests do not support the Authors concept. Obviously the Authors need to use other anxiety tests here, there are quite a few others available.
Reviewer 4 Report
Comments and Suggestions for Authors
Manuscript ID: brainsci-3831252
Title: miR-218-1 regulates mesoaccumbens circuit function and influences behavior in mice
The manuscript investigates the role of miR-218 isoforms in regulating VTA dopamine neurons, dopamine release in the nucleus accumbens, and behavior in mice. The study uses electrophysiology, amperometry, and limited behavioral tests. The main finding is that miR-218-1 deletion causes VTA neuron hyperexcitability and a trend toward anxiety-like behavior, while both isoforms impair dopamine release. This manuscript reports interesting preliminary findings but draws conclusions unsupported by the data. The core paradox (increased firing, decreased release) remains unexplained, behavioral effects are marginal, and sample sizes are insufficient.
Major comments
- Claims that miR-218-1 is the main regulator of mesoaccumbens circuitry are too strong, since both isoforms impaired DA release.
- Missing calibration details for amperometry and exact parameters for reproducibility.
- Different age ranges used (P40-P60) without clear justification for this 20-day window. No counterbalancing mentioned for behavioral testing order. Slice preparation and recording conditions vary between experiments without clear standardization protocols.
- No pharmacological validation that recorded neurons are indeed dopaminergic (e.g., response to dopamine receptor agonists/antagonists). Relies solely on anatomical location and GFP expression for cell identification. No verification that miR-218 deletion specifically affects the intended molecular targets
- Anxiety-like behavior shows only a "tendency" (no significant p-value reported) yet conclusions are drawn as if this were significant. Marble burying test shows no difference, contradicting the anxiety phenotype claim. Need additional anxiety/depression behavioral assays to support conclusions
- KO1 shows hyperexcitability but ALL genotypes show reduced DA release. This paradox is not adequately explained - if KO1 neurons fire more, why is DA release reduced?. The authors speculate about circuit-level effects but provide no evidence
Minor comments
- Ensure consistent abbreviation usage (e.g., c-KO2 vs. cKO2).
- Figure quality could be improved (particularly Figure 1E current traces)
- Methods section lacks sufficient detail for replication of slice preparation and recording protocols
Round 2
Reviewer 2 Report
Comments and Suggestions for Authors
The authors have addressed my comments, and I have no further suggestions.
Author Response
We are grateful to Reviewer for his/her comment to our revised manuscript
Reviewer 3 Report
Comments and Suggestions for Authors
Dear Authors,
I accept your respond provided for my review and believe the revised paper is suitable for publication in the Journal.
Author Response

(The authors gave the same response as above.)

Reviewer 4 Report
Comments and Suggestions for Authors
The findings, while technically sound, are incremental extensions of the authors’ previous work, with limited novelty and lacking mechanistic depth. Moreover, the behavioral data are weak, showing only a non-significant trend that does not convincingly support the conclusions. I therefore recommend rejection.
Author Response
We are grateful to the Reviewer for commenting our manuscript. We are sorry if he/she does not recommend it for publication.